# New idtracker.ai rethinks multi-animal tracking as a representation learning problem to increase accuracy and reduce tracking time

**Jordi Torrents[†], Tiago Costa[†], Gonzalo de Polavieja***

Champalimaud Research, Champalimaud Center for the Unknown, Lisbon, Portugal

## eLife Assessment

This **important** study introduces an advance in multi-animal tracking by reframing identity assignment as a self-supervised contrastive representation learning problem. It eliminates the need for segments of video where all animals are simultaneously visible and individually identifiable, and significantly improves tracking speed, accuracy, and robustness with respect to occlusion. This innovation, which is supported through **compelling** evidence, has implications beyond animal tracking, potentially connecting with advances in behavioral analysis and computer vision.

## Abstract

idTracker and idtracker.ai approach multi-animal tracking from video as an image classification problem. For this classification, both rely on segments of video where all animals are visible to extract images and their identity labels. When these segments are too short, tracking can become slow and inaccurate and, if they are absent, tracking is impossible. Here, we introduce a new idtracker.ai that reframes multi-animal tracking as a representation learning problem rather than a classification task. Specifically, we apply contrastive learning to image pairs that, based on video structure, are known to belong to the same or different identities. This approach maps animal images into a representation space where they cluster by animal identity. As a result, the new idtracker.ai eliminates the need for video segments with all animals visible, is more accurate, and tracks up to 700 times faster.

## Introduction

Video-tracking systems that attempt to follow individuals frame-by-frame can fail during occlusions, resulting in identity swaps that accumulate over time (*Branson et al., 2009*; *Plum, 2024*; *Chen et al., 2023*; *Chiara and Kim, 2023*; *Liu et al., 2023*; *Bernardes et al., 2021*). idTracker (*Pérez-Escudero et al., 2014*) introduced the paradigm of animal tracking by identification from the animal images. This approach, unfeasible for humans, avoids the accumulation of errors by identity swaps during occlusions. Its successor, idtracker.ai *Romero-Ferrero et al., 2019*, built on this paradigm by incorporating deep learning and achieved accuracies often exceeding 99.9% in videos of up to 100 animals.

The core idea of the original idtracker.ai is to use a segment of the video in which all animals are visible to extract a set of images and identity labels for each individual. These labeled images are used to train a convolutional neural network (CNN) with one class per animal. Once trained, the network assigns identities to other segments in which all animals are visible. Only segments with identity assignments that meet strict quality criteria are retained, and their images and labels are also used for

*For correspondence:
gonzalo.polavieja@neuro.
fchampalimaud.org

[†]These authors contributed equally to this work

Competing interest: The authors declare that no competing interests exist.

further training of the CNN. This iterative process of training, assigning, and selecting continues until most of the animal images in the video have been assigned to identities.

If no segment exists in which all animals are visible, the original idtracker.ai cannot start. More commonly, such segments can be short and result in a CNN of low quality. To improve performance in this case, the original idtracker.ai pretrains the CNN using the entire video, but this process is slow. As a consequence, when the segments in which all animals are visible are too short, accuracy might be lower and tracking time longer.

## Results

We built a benchmark to test the original version of idtracker.ai against the new ones. We quantified tracking accuracy using the standard Identification F1 Score (IDF1) (see Appendix 1). We rely on IDF1 because it reflects whether trajectories preserve correct identity assignments throughout the video. *Figure 1a* (blue line) gives the median IDF1 scores for the original idtracker.ai in our benchmark of 33 videos. The first 15 videos of the benchmark are videos of zebrafish, flies (*Drosophila*), and mice for which the original idtracker.ai has an IDF1 score of >99.9%. In the remaining videos, the IDF1 score decreases, reaching 93.77% in video $m\_4\_2$, and 69.66% in video $d\_100\_3$, which lies outside the plotted range.

These accuracies correspond to all animal images in the video excluding images where animals cross paths. We isolate this metric because the core identification algorithm, which is the primary novelty presented in this article, works on these single-animal images to track identities despite the thousands of crossings typically existing in each video. Following this identification, idtracker.ai applies a deterministic post-processing pipeline. This pipeline includes the correction of misidentifications by enforcing the same identity to all individual images between two animal crossings and by detecting impossible abrupt changes in location. It also includes computing animal positions during crossings using the predicted identities from immediately before and after the crossing. This post-processing pipeline remains unchanged from the original idTracker publication (*Pérez-Escudero et al., 2014*), where its details are fully described. We also report the accuracy as IDF1 scores for the complete video including crossings in *Figure 1—figure supplement 1* as this is the final accuracy the user experiences. Further details regarding the benchmark are available in Appendix 1.

*Figure 1b* (blue line) shows the median times that the original idtracker.ai takes to track each of the videos in the benchmark. Some of the videos take a few minutes to track, others a few hours, and six videos take more than three days, one nearly two weeks.

We learn from the benchmark of the original idtracker.ai that it fails to accurately track animals in challenging videos, and its computational time can be of days or weeks, in practice a bottleneck in the study of group behavior from video.

## Optimizing idtracker.ai without changes in the learning method

We first optimized idtracker.ai without changing how we identify animals. We improved data loading and redesigned the main objects in the software (see Appendix 2 for details). This version of the optimized original idtracker.ai (version 5 of the software) achieved higher accuracies, *Figure 1a* (orange line), and *Figure 1—figure supplement 1a* (orange line) for results that include animal crossings. The mean IDF1 score across the benchmark is 99.63% without crossings and 99.49% with crossings, compared with 98.39% without crossings and 98.24% with crossings for the original idtracker.ai (see *Figure 1—figure supplement 2a and b* for boxplots showing more statistics).

Tracking times were also significantly reduced. As shown in *Figure 1b* (orange line), no video took longer than a day. On average, tracking is 13.5 times faster than with the original version and 120.1 times faster for the more difficult videos (see *Figure 1—figure supplement 2c* for boxplots showing more statistics). Despite the large improvement in tracking times, some of them are many hours or close to a day, which would still be too limiting in many pipelines.

To further improve both accuracy and tracking speed, we retained these optimizations while changing the core identification algorithm to eliminate the need for segments of video in which all animals are visible.

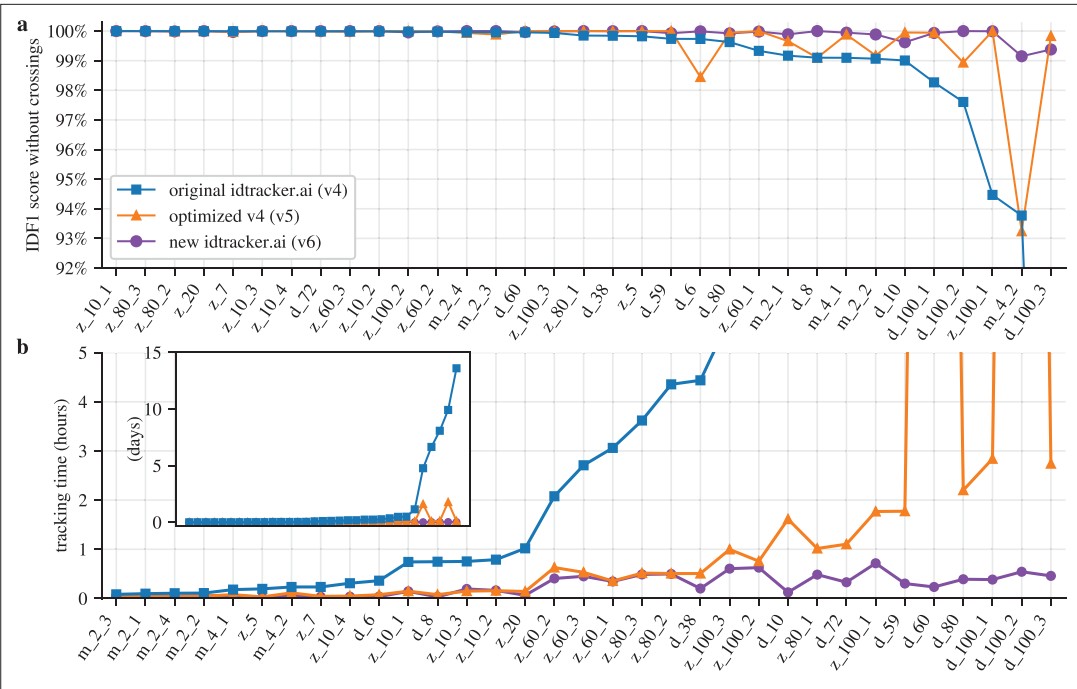

**Figure 1.** Performance for a benchmark of 33 videos of flies, zebrafish, and mice. (**a**) Median IDF1 score computed using all images of animals in the videos excluding animal crossings. The videos are ordered by decreasing IDF1 score of the original idtracker.ai results for ease of visualization. (**b**) Median tracking times are shown for the scale of hours and, in the inset, for the scale of days. The videos are ordered by increasing tracking times in the original idtracker.ai results for ease of visualization. The names of the videos in (**a**) and (**b**) start with a letter for the species ($z$,$d$,$m$), followed by the number of animals in the video, and possibly an extra number to distinguish the video if there are several of the same species and animal group size. Lines between points are for visualization purposes only.

The online version of this article includes the following source data and figure supplement(s) for figure 1:

**Source data 1.** Numerical benchmark results with median, mean, and 20–80 percentile values of tracking accuracy and times.

**Figure supplement 1.** Performance for the benchmark with full trajectories with animal crossings.

**Figure supplement 2.** Boxplot representation of the benchmark results.

**Figure supplement 3.** Robustness to blurring and light conditions.

**Figure supplement 4.** Memory usage across the different software.

## The new idtracker.ai uses representation learning

We reformulate multi-animal tracking as a representation learning problem. In representation learning, we learn a transformation of the input data that makes it easier to perform downstream tasks (*Xing et al., 2002*; *Bengio et al., 2013*; *Ericsson et al., 2022*). In our case, the downstream task is to cluster images into animal identities without requiring identity labels.

This reformulation is made possible by the inherent structure of the video, illustrated in *Figure 2a*. First, we detect specific moments in the video when animals touch or cross paths. These are shown in *Figure 2a* as boxes with dashed borders that contain images of overlapping animals. We can then divide the rest of the video into individual fragments, each consisting of the set of images of a single individual between two animal crossings. *Figure 2a* shows 14 such fragments as rectangles with a gray background. In addition, a video with $N$ animals may contain global fragments, that is, a collection of $N$ individual fragments that coexist in one or more consecutive frames. An example is shown in *Figure 2a* by the five fragments with blue borders. These global fragments are used by the original idtracker.ai to train a CNN. In the new approach, we do not assume that global fragments exist.

Representation learning becomes feasible in this context because we can obtain both positive and negative image pairs without using identity labels. Positive pairs are images of the same individual

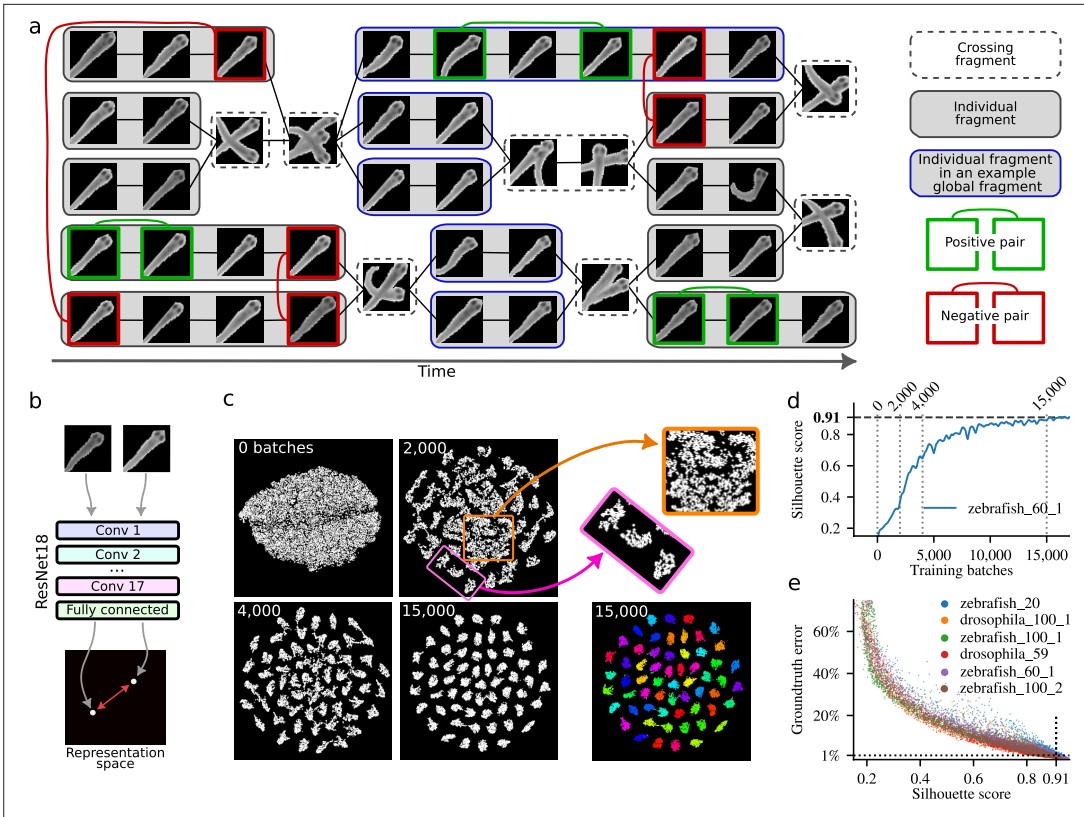

**Figure 2.** Tracking by identification using deep contrastive learning. (**a**) Schematic representation of a video with five fish. (**b**) A ResNet18 network with eight outputs generates a representation of each animal image as a point in an eight-dimensional space (here shown in 2D for visualization). Each pair of images corresponds to two points in this space, separated by a Euclidean distance. The ResNet18 network is trained to minimize this distance for positive pairs and maximize it for negative pairs. (**c**) 2D t-SNE visualizations of the learned 8-dimensional representation space. Each dot represents an image of an animal from the video. As training progresses, clusters corresponding to individual animals become clearer. Here, we plot this process for the example video *zebrafish_60_1* after training for 0, 2000, 4000, and 15,000 batches (each batch contains 400 positive and 400 negative pairs of images, that is, 1600 images per batch). The t-SNE plot at 15,000 training batches is also shown color-coded by human-validated ground-truth identities. The pink rectangle at 2000 batches of training highlights clear clusters, and the orange square fuzzy clusters. (**d**) The Silhouette score measures cluster coherence and increases during training, as illustrated for a video with 60 zebrafish. (**e**) A Silhouette score of 0.91 corresponds to a human-validated error rate of less than 1% per image.

obtained from within the same individual fragment (*Figure 2a*, green boxes). Negative pairs are images of different individuals taken from different individual fragments that coexist in time for one or more frames (*Figure 2a*, red boxes). We can then use the positive and negative pairs of images for contrastive learning, a self-supervised learning framework designed to learn a representation space in which positive examples are close together, and negative examples are far apart (*Schroff et al., 2015*; *Dong and Shen, 2018*; *Kaya and Bilge, 2019*; *Chen et al., 2020a*; *Chen et al., 2020b*; *Guo et al., 2020*; *Wang et al., 2020*; *Yang et al., 2020*). This formulation is supported by the success of deep metric and contrastive learning methods (see Appendix 3 for comparison with previous work).

We first evaluated which neural networks are suitable for contrastive learning with animal images. In addition to our previous CNN from idtracker.ai, we tested 31 networks from 10 different families of state-of-the-art CNNs and transformer architectures, selected for their compatibility with consumer-grade GPUs and ability to handle small input images (20×20 to 100×100 pixels) typical in collective animal behavior videos. Among these architectures, ResNet18 (v1) (*He et al., 2016a*) without pretrained weights was the fastest to obtain low errors (see Appendix 3).

A ResNet18 with $M$ outputs maps each input image to a point in an $M$-dimensional representation space (illustrated in *Figure 2b* as a point on a plane). Experiments showed that using $M = 8$ achieved

faster convergence to low error (see Appendix 3). ResNet18 is trained using a contrastive loss function (*Chopra et al., 2005*, see Appendix 3 for details). Each image in a positive or negative pair is input separately into the network, producing a point in the eight-dimensional representation space. For an image pair, we then obtain two points in an eight-dimensional space, separated by some (Euclidean) distance. The optimization of the loss function minimizes (or maximizes) this Euclidean distance for positive (or negative) pairs until the distance $D_{pos}$ (or $D_{neg}$) is reached. The effect of $D_{pos}$ is to prevent the collapse to a single of the positive images coming from the same fragment, allowing for a small region of the eight-dimensional representation space to contain all the positive pairs of the same identity. The effect of $D_{neg}$ is to prevent excessive scatter of the points representing images from negative pairs. We empirically determined that $D_{neg}/D_{pos} = 10$ results in a faster method to obtain low error (see Appendix 3), and we use $D_{pos} = 1$ and $D_{neg} = 10$.

As the model trains, the representation space becomes increasingly structured, with similar data points forming coherent clusters. *Figure 2c* visualizes this progression using 2D t-SNE (*Maaten and Hinton, 2008*) plots of the eight-dimensional representation space. After 2000 training batches (400 positive and 400 negative pairs of images per batch), initial clusters emerge, and by 15,000 batches, distinct clusters corresponding to individual animals are evident. Ground truth identities verified by humans confirm that each cluster corresponds to an animal identity (*Figure 2c*, colored clusters).

The method to select positive and negative pairs is critical for fast learning (*Awasthi et al., 2022*; *Khosla et al., 2021*; *Rösch et al., 2024*). This is because not all image pairs contribute equally to training. *Figure 2c* shows at 2000 training batches that some clusters are well-defined (e.g., those inside the pink rectangle) while others remain fuzzy (e.g., those inside the orange square). Images in well-defined clusters have negligible impact on the loss or weight updates, as positive pairs are already close and negative pairs are sufficiently separated. Sampling from these well-defined clusters, therefore, wastes time. In contrast, fuzzy clusters contain images that still contribute significantly to the loss and benefit from further training. To address this, we developed a sampling method that prioritizes pairs from underperforming clusters requiring additional learning, while maintaining baseline sampling for all clusters based on fragment size (see Appendix 3). This ensures consistent updates across the representation space and prevents forgetting in well-defined clusters.

To assign identities to animal images, we perform k-means clustering (*Sculley, 2010*) on the points representing all images of the video in the learned eight-dimensional representation space. Each image is then assigned to a cluster with a probability that increases the closer it is to the cluster center. To evaluate clustering quality, we compute the mean Silhouette score (*Rousseeuw, 1987*), which quantifies intra-cluster cohesion and inter-cluster separation. A maximum value of 1 indicates ideal clustering. During training, the mean Silhouette score increases (*Figure 2d*). We empirically determined that a value of 0.91 for this index corresponds to an identity assignment error below 1% for a single image (*Figure 2e*). As a result, we use 0.91 as the stopping criterion for training (see Appendix 3).

The new idtracker.ai (v6) is more accurate than original idtracker.ai (v4) and than its optimized version (v5), *Figure 1a* (purple line). Its average IDF1 score in the benchmark is 99.92% and 99.82% without and with crossings, respectively, an important improvement over the original idtracker.ai v4 (98.39% and 98.24%) and its optimized version v5 (99.63% and 99.49%). It also gives much shorter times than the original idtracker.ai (v4) and its optimized version (v5), *Figure 1b* (purple line). It is, on average, 90 times faster than the original idtracker.ai (v4) and, for the more difficult videos, up to 712 times faster. See *Figure 1—figure supplement 2* for boxplots showing more statistics comparing tracking systems.

As for the original idtracker.ai, the new idtracker.ai can work well with lower resolutions, blur and video compression, and with inhomogeneous light (*Figure 1—figure supplement 3*). We also compared the new idtracker.ai to TRex (*Walter and Couzin, 2021*), which is based on idtracker.ai without pretraining and with additional operations like eroding crossings to make global fragments longer, posture image normalization, tracklet subsampling, and the use of uniqueness feedback during training. TRex gives comparable accuracies to the original idtracker.ai in the benchmark, and it is on average 31 times faster than the original idtracker.ai and up to 316 times faster (*Figure 1—figure supplement 1b*). However, the new idtracker.ai is both more accurate and faster than TRex (*Figure 1—figure supplement 1*). The mean IDF1 score of TRex across the benchmark is 98.14% and 97.89% excluding and including animal crossings, respectively. This is noticeably below the values for

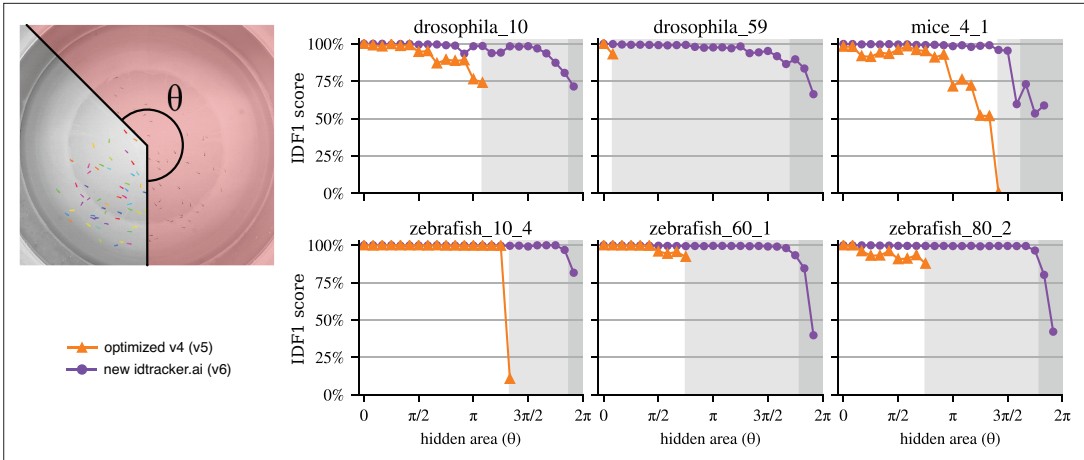

**Figure 3.** Tracking with strong occlusions. Accuracies when we mask a region of a video defined by an angle $\theta$ and the tracking system has no access to the information behind the mask. Light and dark gray regions correspond to the angles for which no global fragments exist in the video. Dark gray regions correspond to angles for which the video has a fragment connectivity lower than 0.5, with the fragment connectivity defined as the average number of other fragments each fragment coexists with, divided by $N - 1$, with $N$ the total number of animals; see *Figure 3—figure supplement 1*, for an analysis justifying this value of 0.5 . The original idtracker.ai (v4) and its optimized version (v5) cannot work in the gray regions, and new idtracker.ai is expected to deteriorate only in the dark gray region.

The online version of this article includes the following figure supplement(s) for figure 3:

**Figure supplement 1.** Fragment connectivity analysis.

the new idtracker.ai of 99.92% and 99.82%, respectively. Also, the new idtracker.ai is on average 4.5 times faster and up to 16.5 times faster than TRex. See *Figure 1—figure supplement 2* for boxplots showing more statistics for IDF1 scores and tracking times. Additionally, the new idtracker.ai has a memory peak lower than TRex (*Figure 1—figure supplement 4*).

## No need for global fragments

The new idtracker.ai also works in videos in which the original idtracker.ai does not even track because there are no global fragments. Global fragments are absent in videos with very extensive animal occlusions, for example, because animals touch or cross more frequently, parts of the setup are covered, or the camera focuses on only a specific region of the setup.

To systematically evaluate this, we tracked videos in which we erased sectors of angle θ (*Figure 3*). These sectors are painted black before the tracking starts so animals inside this region are not visible to the tracking systems (see 'Methods' for more details about the occlusion tests).

The light and dark gray regions in *Figure 3* correspond to videos with no global fragments, and the original idtracker.ai and its optimized version declare tracking impossible in these regions. The new idtracker.ai, however, works well until approximately 1/4 of the setup is visible, and afterward, it degrades. This shows the limit of the new idtracker.ai. The deterioration happens because, for the clustering process to be successful, we need enough coexisting individual fragments to have both positive and negative examples. We measure this using fragment connectivity, defined as the average number of other fragments a fragment coexists with, divided by $N - 1$, with $N$ the total number of animals in the video. Empirically, we find that a fragment connectivity below 0.5 corresponds to low accuracies (*Figure 3—figure supplement 1*). The new idtracker.ai warns the user when this condition of low fragment connectivity takes place, which we indicate in *Figure 3* with the dark gray regions.

## Output of new idtracker.ai

The final output of the new idtracker.ai consists of the $x$ and $y$ coordinates for each identified animal and video frame. Additionally, it provides an estimation of the achieved accuracy, the Silhouette score as a measure of clustering quality, and the probability of correct identity assignment for each animal

and frame. The new idtracker.ai also includes the following tools, for which we also give an example workflow in Appendix 4 and documentation at https://idtracker.ai/latest/user_guide/tools.html:

- `idtrackerai_inspect_clusters`, a tool to visually inspect the learned representation space and check the clusters integrity.
- `idtrackerai_validate`, a graphic app to review and correct tracking results.
- `idtrackerai_video`, a graphic app to generate videos of the computed animal trajectories overlaid on the original video for visualization. This app also generates individual videos for each animal showing only its cropped region over time to be able to run pose estimators like the ones in *Lauer et al., 2022*; *Pereira et al., 2022*; *Segalin et al., 2021*; *Tang et al., 2025*; and *Biderman et al., 2024*.
- `idmatcher.ai`, a tool to match identities across multiple recordings (see Appendix 5).
- Direct integration with SocialNet, a model of collective behavior introduced in *Heras et al., 2019*.
- Direct integration with *trajectorytools*, a Python package for 2D trajectory processing, and a set of Jupyter Notebooks that uses *trajectorytools* to analyze basic movement properties and spatial relationships between the animals.

## Discussion

We have introduced a new way to perform multi-animal tracking that shifts from a classification task to a representation learning one. By leveraging contrastive learning on image pairs derived from the temporal structure of the video, the new idtracker.ai eliminates the restrictive requirement for segments where all animals are simultaneously visible. The idea of contrastive identification could also be of value in other contexts, for example, in tracking body parts.

The new idtracker.ai is not only more robust to occlusions but also outperforms previous tracking systems both in accuracy and tracking time. With a median IDF1 score of 99.92% and processing speeds up to 700 times faster than previous versions, the software transforms tracking from a computational bottleneck requiring days or weeks of processing into a much faster step suitable for agile experimental loops.

This approach expands the scope of feasible behavioral studies, allowing for more complex environments where animals are frequently occluded or move in and out of the field of view. Our experiments with masked video regions demonstrate that the system requires only a minimum sufficient pairwise co-occurrence of animals rather than complete group visibility, making it robust to conditions where previous algorithms fail.

Additionally, we have surrounded this new core algorithm with a software ecosystem to improve user experience. We introduced interactive tools that allow researchers to visually inspect the learned clusters and easily validate trajectories. We also prioritized interoperability by facilitating the generation of data for pose-estimation frameworks like DeepLabCut (*Lauer et al., 2022*) and SLEAP (*Pereira et al., 2022*) for fine-grained body analysis, and the direct integration with modeling of group behavior like SocialNet (*Heras et al., 2019*).

## Methods
### Software availability

idtracker.ai is a free and open-source project (license GPLv3). Information about its installation and usage can be found on the website https://idtracker.ai/. The source code is available in https://gitlab.com/polavieja_lab/idtrackerai (*Torrents et al., 2026*) and the package is pip-installable from PyPI. All versions can be found in these platforms, specifically '*original idtracker.ai (v4)*' as v4.0.12, '*optimized v4 (v5)*' as v5.2.12 and '*new idtracker.ai (v6)*' as v6.0.9. We only actively maintain and provide support for the latest version available, having the old ones for archive and reference only.

### Tested computer specifications

The software idtracker.ai depends on PyTorch and is thus compatible with any machine that can run PyTorch, including Windows, MacOS, and Linux systems. Although no specific hardware is required, a graphics card is highly recommended for hardware-accelerated machine-learning computations.

Version 6 of idtracker.ai was tested on computers running Ubuntu 24.04, Fedora 41, Windows 11 with NVIDIA GPUs from the 1000 to the 4000 series, and MacOS 15 with Metal chips. The benchmark results presented in this study were performed on a desktop computer running Ubuntu 24.04 LTS 64 bit with an AMD Ryzen 9 5950X (32 cores at 3.4 GHz) processor, 128 GB RAM, and an NVIDIA GeForce RTX 4090.

## Occlusion tests

We clarify here some details from the occlusion tests presented in the section 'No need for global fragments'. The occlusion tests were performed using the same set of videos as in the benchmark. For these tests, we defined an occlusion mask as a region of interest in the software. When video frames are converted into binary foreground-background images, all pixels inside the mask are treated as background, so no information is extracted from them.

The fragments are detected as in any other video. Specifically, this means that when one animal is hidden in the mask, the animal is lost and the fragment breaks. No hidden visual information outside the region of interest is used to track the visible portion, so fragments are not built adding any artificial links.

For evaluation, the ground truth of the unmasked video containing the positions of all animals at all times was used, but, obviously, only positions outside the mask were used to compute tracking accuracy. To avoid partial detections of the animals, trajectories within 15 pixels of the mask boundary (75 pixels for mice videos) were excluded from the evaluation.

## Acknowledgements

We thank Alfonso Perez-Escudero, Paco Romero-Ferrero, Francisco J Hernandez Heras, Madalena Valente for discussions, and three anonymous reviewers for many suggestions. This work was supported by Champalimaud Foundation and by Fundação para a Ciência e Tecnologia in the context of projects PTDC/BIA-COM/5770/2020 and UID/04443/2025.

## Additional information

### Funding

| Funder | Grant reference number | Author |
|---|---|---|
| Fundação para a Ciência e Tecnologia | PTDC/BIA-COM/5770/2020 | Gonzalo de Polavieja |
| Fundação para a Ciência e Tecnologia | UID/04443/2025 | Gonzalo de Polavieja |

The funders had no role in study design, data collection and interpretation, or the decision to submit the work for publication.

### Author contributions

Jordi Torrents, Resources, Data curation, Software, Validation, Investigation, Methodology, Writing – review and editing; Tiago Costa, Conceptualization, Formal analysis, Investigation, Methodology, Writing – review and editing; Gonzalo de Polavieja, Conceptualization, Formal analysis, Supervision, Funding acquisition, Methodology, Writing – original draft, Project administration, Writing – review and editing

### Author ORCIDs

Jordi Torrents https://orcid.org/0009-0006-6353-4079
Tiago Costa https://orcid.org/0000-0002-8538-1345
Gonzalo de Polavieja https://orcid.org/0000-0001-5359-3426

Reviewer #3 (Public review): https://doi.org/10.7554/eLife.107602.4.sa1
Author response https://doi.org/10.7554/eLife.107602.4.sa2

# Additional files

## Supplementary files
MDAR checklist

## Data availability
All videos used in this study, their tracking parameters and human-validated groundtruth can be found in our data repository at https://idtracker.ai/.

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

# Appendix 1

## Computation of tracking accuracy

We used the idtracker.ai Validator tool (see Appendix 4) to manually generate ground-truth trajectories. The input to the Validator were outputs from idtracker.ai v5, and the Validator facilitates that the user manually corrects the mistakes. The obtained ground truth includes the position and identity of each animal in every frame, along with their classification as either individual or crossing.

Tracking accuracy is quantified using the standard Identification F1 Score (IDF1), defined in *Ristani et al., 2016* as:

$$\text{IDF1} = \frac{2\text{TP}}{2\text{TP} + \text{FP} + \text{FN}} \tag{1}$$

Here, TP (true positive) refers to predicted positions that match their ground-truth points within a distance $T$. FP (false positive) are predicted positions with no corresponding ground-truth match within a distance $T$, and FN (false negative) are ground-truth positions with no matching prediction within a distance $T$.

An identity switch in a single frame results in two false positives and two false negatives, as both predicted and ground-truth identities become unmatched. False negatives also occur when the software either fails to detect an animal or loses track of its identity in a given frame.

We rely on IDF1 rather than other multi-object tracking metrics such as Multiple Object Tracking Accuracy (MOTA) or Multiple Object Tracking Precision (MOTP) (*Bernardin and Stiefelhagen, 2008*), since the central goal of our system is to maintain consistent identities across time. MOTA counts an identity switch as a single error regardless of its persistence, thereby underestimating the severity of prolonged misidentifications. MOTP evaluates how well predicted positions or bounding boxes align with ground-truth locations, focusing on spatial localization accuracy. While useful for detection-centric benchmarks, neither MOTA nor MOTP adequately capture the temporal consistency of identities. In contrast, IDF1 directly reflects whether trajectories preserve correct identity assignments throughout the video, which is the relevant criterion for evaluating our system.

We report two types of accuracy: accuracy with crossings, which includes all trajectory points; and accuracy without crossings, where points labeled as crossings in the ground truth are excluded from the evaluation.

We present all results using $T = 1\text{BL}$ with BL being a body length. We also verified that accuracy remains largely unaffected by the value of $T$. For instance, reducing it to $T = 0.5\text{BL}$ results in a very small change of the mean IDF1 score (without crossings) across the benchmark in the new idtracker. ai from 99.923% to 99.908%.

## Benchmark of accuracy and tracking time

To evaluate the tracking time and accuracy of versions 4 (4.0.12), 5 (5.2.12), and 6 (6.0.9) of idtracker. ai and version 1.1.9 of TRex, we used a set of 33 videos with their corresponding human-validated ground-truth trajectories. Each video is 10 minutes long, has a frame rate between 25 and 60 fps, and features one of three species: mice, *Drosophila*, or zebrafish, with the number of individuals ranging from 2 to 100. Crossing blobs in these videos make up, on average, 2.6% of the total amount of blobs (1.1% for zebrafish, 0.7% for *Drosophila*, and 9.4% for mice videos).

Both idtracker.ai and TRex rely on several tracking parameters. Some of them have default values that typically require no adjustment (e.g., minimum blob size, network and training hyperparameters) or have a single correct value (e.g., number of animals). In contrast, variable parameters do not have a unique correct value but rather a valid range, from which the user must select a specific value to run the system. In idtracker.ai, the only variable parameter is `intensity_threshold`, whereas in TRex, two such parameters exist: `threshold` and `track_max_speed`.

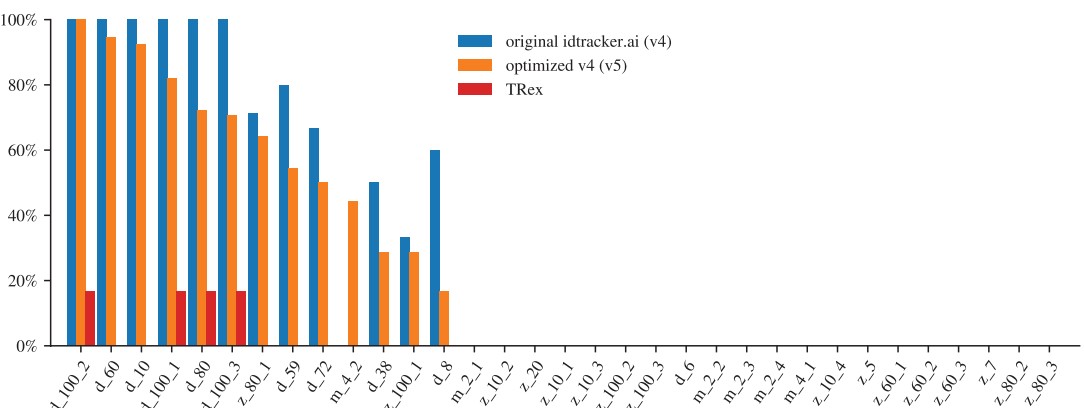

**Appendix 1—figure 1.** Protocol 2 failure rate. Probability for the different tracking systems of not tracking the video with protocol 2 in idtracker.ai (v4 and v5) and in TRex the probability that it fails without generating trajectories.

For the variable parameters, choosing one value or another within the valid interval can give different tracking results. For some values, previous versions of idtracker.ai (v4 and v5) can stochastically fail at tracking with the default protocol 2 and resort to what we call protocol 3 (*Appendix 1—figure 1*), a method that can take days to process more complex videos. Similarly, TRex can crash without outputting any trajectory in certain videos, leading to missing IDF1 score outputs (*Appendix 1—figure 1*).

For this reason, simulating a user that, for a given video, runs the tracking software a single time can result in extremely slow tracking for previous versions of idtracker.ai and a failed tracking for TRex. This would give an advantage to the new version v6 of idtracker.ai since it does not have the slow protocol 3 and never crashed in our tests. We thus simulated a more realistic user that, out of up to 5 attempts, uses the first run in which there is no use of the slow protocol 3 in idtracker.ai v4 and v5 (or protocol 3 is used if it consumes the 5 runs) and in TRex the first run in which the software does not crash. The tracking times are then the sum of tracking times of the attempts used.

To simulate our user many times, we first prepared the dataset of tracking runs by repeating the tracking for each video and software until reaching 5 successful runs or a maximum of 35 attempts. For the original version of idtracker.ai, this was limited to 3 successful runs or 7 attempts due to significantly longer tracking times. Each new run uses new random values for the variable parameters sampled from a precomputed interval we consider an expert user can choose from. In successful runs, both IDF1 score and tracking time are recorded. In failed runs, when idtracker.ai switches to protocol 3 or TRex crashes, only the time until failure is recorded.

We then simulated our user up to 10,000 times per software and video by sampling tracking runs from our dataset of tracking runs to obtain robust estimates of the tracking times and accuracies. *Figures 1* and *Figure 1—figure supplement 1* report the median accuracies, without and with crossings, respectively, and tracking times. The source data is presented in *Figure 1—source data 1* with the median, mean, and the 20 and 80 percentiles values.

To ensure a fair comparison, TGrabs (a video pre-processing tool needed by TRex) is included when running TRex, graphical interfaces are always disabled at runtime to maximize performance, `output_interpolate_positions` is enabled in TRex and posture estimation was not deactivated nor used in TRex.

## Appendix 2

### Improvements to the original idtracker.ai in version 5

Following the last publication of idtracker.ai *Romero-Ferrero et al., 2019*, the software underwent continuous maintenance, including feature additions, performance optimizations, and hyperparameter tuning (released via PyPI from March 2023 for v5.0.0 to June 2024 for v5.2.12). These updates improved the implementation and tracking pipeline but did not alter the core algorithm. Significant advancements were made in user experience, tool availability, processing speed, and memory efficiency. Below, we summarize the most notable changes.

#### Blob memory optimization

Blobs are defined as collections of connected pixels belonging to one or more animals. In v4, blobs stored pixel indices, causing memory usage to scale quadratically with blob size. In v5, blobs are represented by simplified contours using the Teh-Chin chain approximation (*Teh and Chin, 1989*), reducing memory usage by 93% in blob instances. This also accelerated blob-related computations (centroid, orientation, area, overlap, identification image creation, etc.).

#### Efficient image loading

Identification images are now efficiently loaded on demand from HDF5 files, eliminating the need to load all images into memory. This enables training with all images regardless of video length, with minimal memory usage.

#### Code optimization

The source code was revised to eliminate speed bottlenecks. The most impactful changes include
- Frame segmentation accelerated by 80% through optimized OpenCV usage.
- Faster blob-to-blob overlap checks by first evaluating bounding boxes before deeper comparisons.
- Persistent storage of blob overlap checks to avoid redundant computations when reloading data.
- Efficient disk access for identification images by reading them in sorted batches, minimizing I/O overhead.
- Reduced bounding box image sizes to the minimum necessary, lowering memory and processing demands.
- Optimized and parallelized Torch data loaders for more efficient model training.
- Caching of computationally expensive properties for blobs, fragments, and global fragments.
- Sorted fragment lists to speed up coexistence detection.

#### Changes to the identification protocol

In v4, identity assignments to high-confidence fragments were fixed and excluded from downstream correction, regardless of later evidence. In v5, this was relaxed for short fragments (fewer than four frames), allowing corrections due to their statistical unreliability and frequent image noise.

#### Expanded idtracker.ai ecosystem

A restructure of the main graphic user interface and the creation of a new validation app for inspecting and correcting tracking results. Direct integration of idmatcher.ai, originally introduced in *Romero-Ferrero et al., 2023*, that allows propagation of consistent identity labels across multiple recordings. See Appendix 4 for a full guide on the current extra features of idtracker.ai.

# Appendix 3

We direct the reader to Appendix B of *Romero-Ferrero et al., 2019* for the formal definitions of the main objects and algorithms used in old idtracker.ai, which are common to versions 4, 5, and 6:

- B.2.1 Segmentation
- B.2.2 Detection of individual and crossing images
- B.2.3 Fragmentation
- B.2.5 Residual identification
- B.2.6 Post-processing
- B.2.7 Output

The following sections describe in detail the new identification algorithm introduced in the new idtracker.ai (v6).

## Network architecture

Training represents 47% of the total tracking time in our benchmark. To identify the fastest architecture in training, we evaluated 31 models from 10 families of state-of-the-art CNNs and vision transformers, including the CNN used in versions 4 and 5 of idtracker.ai.

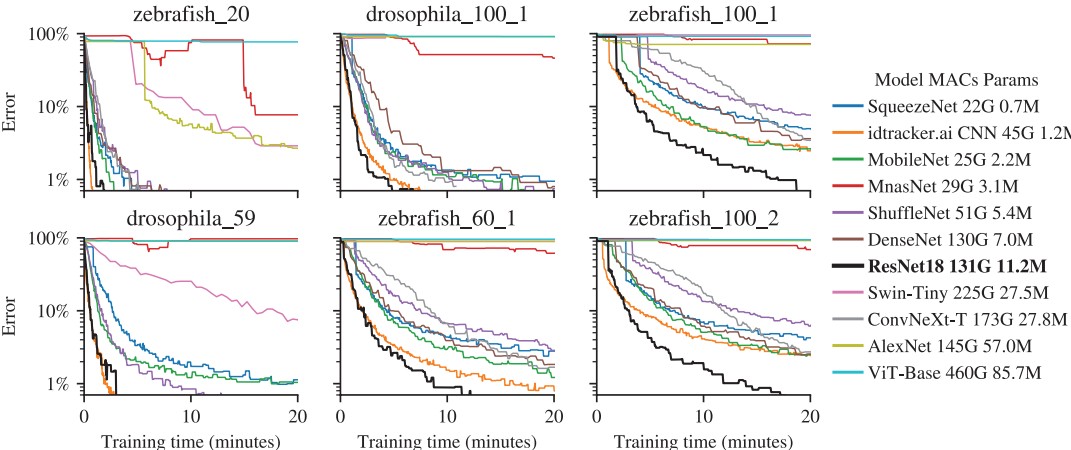

**Appendix 3—figure 1.** Models comparison. Error in image identification as a function of training time for different deep learning models randomly initialized in six test videos. For each network, we report the multiply-accumulate operations (MAC) in giga operations (G) (for a batch of 1600 images of size 40 × 40 × 1) and the number of parameters in the units of million parameters (M). Every 100 training batches, we perform k-means clustering on a randomly selected set of 20,000 images, assigning identities based on clusters. We then compute the Silhouette score and ground-truth error on the same set. The reported error corresponds to the model with the best Silhouette score observed up to that point.

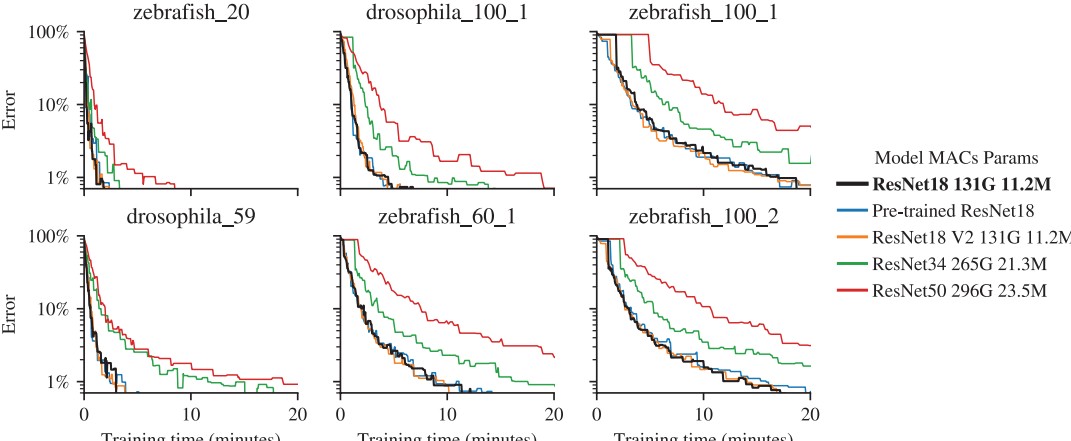

**Appendix 3—figure 2.** ResNet models comparison. Error in image identification as a function of training time for different deep learning models randomly initialized (except *Pre-trained ResNet18*) in six test videos. For each network, we report the multiply-accumulate operations (MAC) in giga operations (G) (for a batch of 1600 images of size 40 × 40 ×1) and the number of parameters in the units of million parameters (M). Every 100 training batches, we perform k-means clustering on a randomly selected set of 20,000 images, assigning identities based on clusters. We then compute the Silhouette score and ground-truth error on the same set. The reported error corresponds to the model with the best Silhouette score observed up to that point.

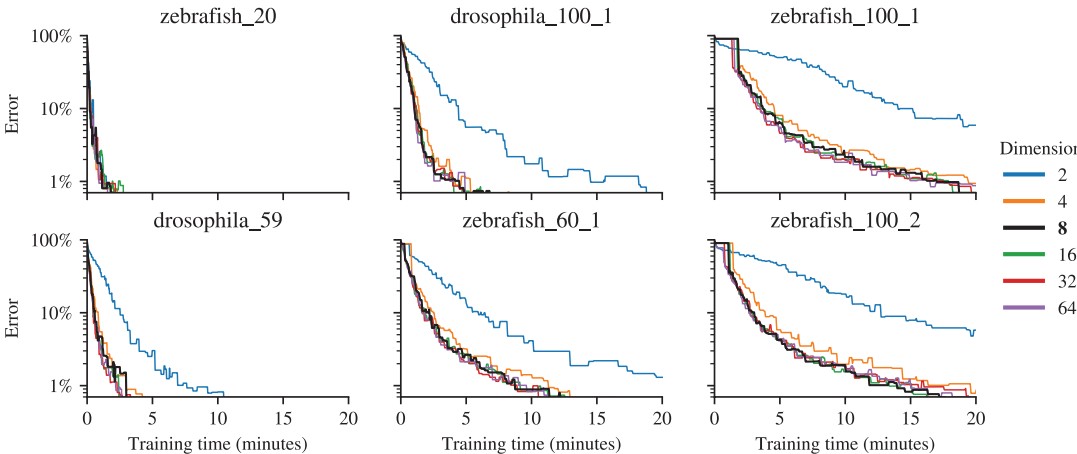

**Appendix 3—figure 3.** Embedding dimensions comparison. Error in image identification as a function of training time for different embedding dimensions in six test videos. Every 100 training batches, we perform k-means clustering on a randomly selected set of 20,000 images, assigning identities based on clusters. We then compute the Silhouette score and ground-truth error on the same set. The reported error corresponds to the model with the best Silhouette score observed up to that point.

Within each family, we tested different network sizes and dropout values and plotted the learning curves of the best candidates (*Appendix 3—figure 1*). The earlier idtracker.ai CNN performed adequately on simpler videos (e.g., *zebrafish_20*) but struggled in more challenging ones (e.g., *zebrafish_100_2*). Vision transformers (ViT *Dosovitskiy et al., 2020*, Swin *Liu et al., 2021*) converged much more slowly, and in some cases could not be trained due to excessive VRAM requirements. Note that image size typically ranges between 20×20 and 100×100 pixels, and each batch consists of 400 positive and 400 negative image pairs (1600 images per batch). We also tested more advanced CNN architectures such as SqueezeNet (*Iandola et al., 2016*), MobileNet (*Howard et al., 2017*), MnasNet (*Tan et al., 2018*), ShuffleNet (*Zhang et al., 2018*), DenseNet (*Huang et al., 2017*), and AlexNet (*Krizhevsky, 2014*). None achieved the same error levels within comparable training times as our baseline CNN. In contrast, ResNet (*He et al., 2016a*) consistently outperformed all other models, offering the best balance between training speed and accuracy across videos.

Within the ResNet family (*Appendix 3—figure 2*), the smallest ResNet18 proved optimal: the largest variants (ResNet101, ResNet152) demanded excessive VRAM and the intermediate sizes (ResNet34 and ResNet50) showed slower convergence. Both v1 and v2 versions (*He et al., 2016b*) performed equivalently. Pre-training on ImageNet conferred no benefit, likely due to the domain mismatch between natural images and video-specific animal crops.

Embedding dimension was another key hyperparameter. Here, too, there is a tradeoff between achieving a robust representation of subtle differences between animals and maintaining a compact network size and efficient training speed. Empirically, an embedding dimension of 8 provided a good trade-off with other values performing similarly (*Appendix 3—figure 3*).

The final *contrastive learning* network (*Figure 2b*) is a ResNet18 (v1) with a single-channel input for grayscale images and an eight-unit fully connected output layer with no bias and using the identity as activation function. Networks are randomly initialized unless the user indicates to copy the weights from a previous tracking session (see Appendix 4 for a usage example). Training is performed with the Adam optimizer *Kingma and Ba, 2017* at a learning rate of 0.001

## Loss function

The contrastive loss function operates on pairs of data points, aiming to minimize the distance between positive pairs and maximize the distance for negative pairs. Being $F_i$ a fragment with arbitrary identifier $i$ and $I_{ik}$ and image in the fragment $F_i$ with arbitrary identifier $k$, the contrastive loss $\mathcal{L}$ for a pair of images $(I_{ik}, I_{jl})$ is defined as:

$$\mathcal{L}(I_{ik}, I_{jl}, l_{ik,jl}) = l_{ik,jl} \cdot \max(0, D_{ik,jl} - D_{\text{pos}})^2$$

$$+ (1 - l_{ik,jl}) \cdot \max(0, D_{\text{neg}} - D_{ik,jl})^2$$

$$l_{ik,jl} = \begin{cases} 1 & \text{if } i = j \text{ (positive pair)} \\ 0 & \text{if } i \neq j \text{ and } F_i \text{ coexists with } F_j \text{ (negative pair)} \end{cases}$$

(2)

where $D_{ik,jl}$ is the Euclidean distance between the embedding of $I_{ik}$ and $I_{jl}$, $D_{\text{neg}}$ is the minimum allowed distance in a negative pair of images (images coming from coexisting fragments), and $D_{\text{pos}}$ is the maximum allowed distance in a positive pair of images (images from the same fragment). It is important to emphasize that the network processes one image at a time, obtaining a single independent point in the representational space for each image. The Euclidean distance between the embeddings for the corresponding pairs of images is computed only afterwards.

$D_{\text{neg}}$ and $D_{\text{pos}}$ serve as thresholds to regulate distances in the embedding space. $D_{\text{neg}}$ prevents images from negative pairs from being pushed indefinitely far apart, while $D_{\text{pos}}$ prevents the collapse of images from positive pairs into a single point.

These thresholds $D_{\text{pos}}$ and $D_{\text{neg}}$ are crucial in our problem, where we aim to embed images of the same identity in similar regions of the representational space. This is necessary because we cannot compare all possible pairs of images and are instead limited to the fragment structure of the video to obtain the labels $l_{ik,jl}$. This limitation means that the loss function does not directly pull together embeddings of the same identity, but rather images from the same fragment. Similarly, the loss does not push apart embeddings of different identities but images from coexisting fragments. $D_{\text{pos}}$ helps prevent the collapse of all images from the same fragment to a single point, allowing for the creation of a diffuse region in the representational space where fragments from the same identity are clustered together. $D_{\text{neg}}$ prevents excessive scattering, ensuring better compression of the representational space and maintaining the integrity of clusters of images from the same identity.

We use $D_{\text{pos}} = 1$ and $D_{\text{neg}} = 10$. These values were determined empirically to provide effective embeddings and were robust for tracking multiple videos across various species and different numbers of animals (*Appendix 3—figure 4*).

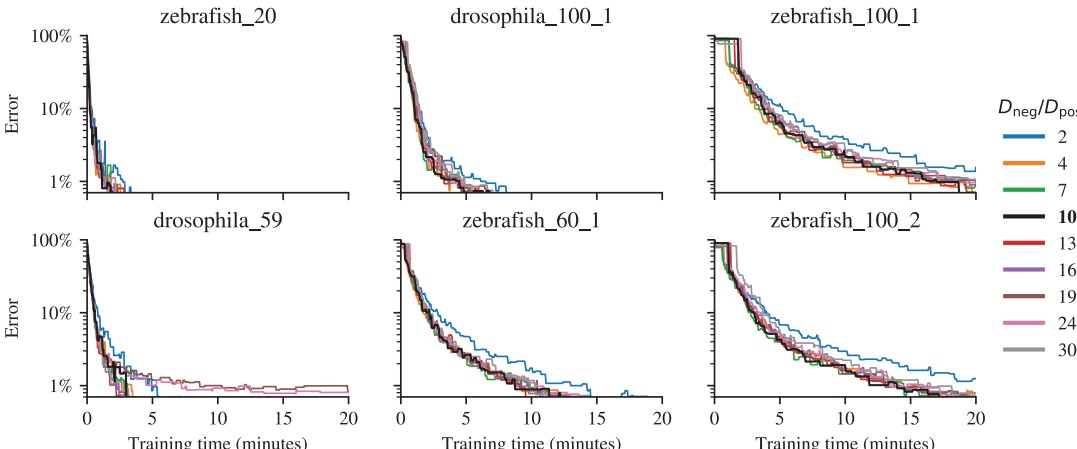

**Appendix 3—figure 4.** $D_{neg}$ over $D_{pos}$ comparison. Error in image identification as a function of training time for different ratios of $D_{neg}/D_{pos}$ in six test videos. Every 100 training batches, we perform k-means clustering on a randomly selected set of 20,000 images, assigning identities based on clusters. We then compute the Silhouette score and ground-truth error on the same set. The reported error corresponds to the model with the best Silhouette score observed up to that point.

## Sampling strategy

Ideally, we would create two datasets of image pairs: one containing negative pairs and another containing positive pairs. However, very long videos or those containing a large number of animals can yield trillions of pairs of images, making the process computationally prohibitive. Therefore, we approach the problem with a hierarchical sampling method. For negative pairs, we first select a pair of coexisting fragments, and then we randomly sample an image from each fragment. For a positive pair, we select a fragment, and then we randomly sample two images from it.

Because of this sampling strategy and the huge size of possible outcomes, the idea of an epoch as a complete pass through the training dataset does not exist here. We instead measure the amount of training by the number of batches used.

Following the hierarchical sampling method, we start by creating two datasets. The first consists of a list of all the fragments in the video, from which we will sample the positive pairs. The second dataset contains all possible pairs of coexisting fragments in the video. From these lists, we exclude all fragments smaller than four images to reduce possible noisy blobs.

Since the same set of images is used both for training and prediction, and these are already pre-aligned egocentrically, we do not apply data augmentation. Augmenting the data would not introduce new useful information and could, in fact, slow down the training process. Therefore, the model is trained directly on the unaltered blob images.

Our tests revealed that large and balanced batches, with an equal number of positive and negative pairs, are ideal for our setting of contrastive learning. Specifically, we choose batches consisting of 400 positive pairs of images and 400 negative pairs of images (1600 images in total), as it was the smallest batch size that did not compromise training speed or accuracy (*Appendix 3—figure 5*). Intuitively, large batch sizes allow for a good spread of pairs from a significant proportion of the video, thereby forcing the network to learn a global embedding of the video. Since positive pairs tend to diminish the size of the representational space while negative pairs tend to increase it, a good balance between the two forces the network to compress the representational space while respecting the negative relationships (*Chen et al., 2020a*). This balance between positive and negative pairs is somewhat surprising, given that several works emphasize the importance of negative examples over positive ones (*Awasthi et al., 2022*; *Khosla et al., 2021*). While we do not yet have an explanation for why this balance appears to perform better in our case, we note that it is not possible to compare all images from one class against those of another, as negative pairs of images can only be sampled from coexisting fragments. Additionally, positive pairs that compress the space can only be sampled from the same fragmetnt and not the same identity. Since we cannot compare images freely and are constrained by the fragment structure of the video, we might need

more positive pairs to ensure a higher degree of compression of the representational space, such that not only images from the same fragment are close together, but also images from the same identity.

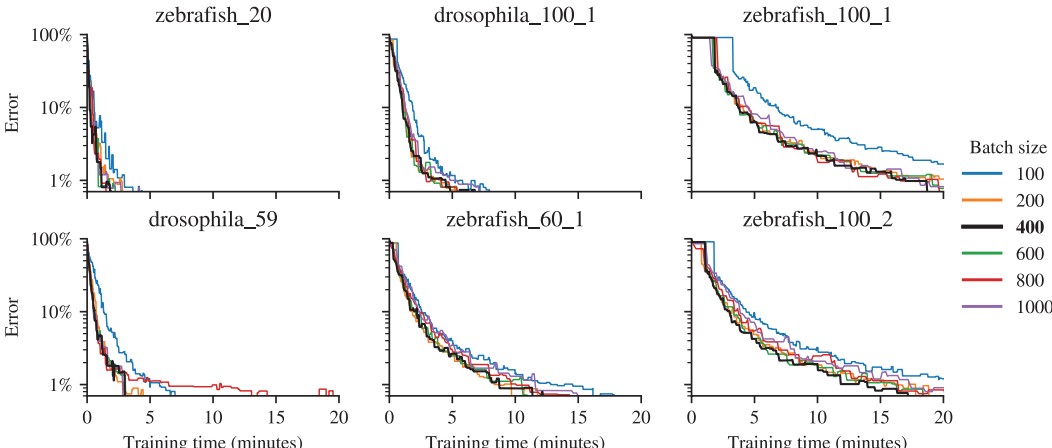

**Appendix 3—figure 5.** Batch size comparison. Error in image identification as a function of training time for different batch sizes of pairs of images in six test videos. Every 100 training batches, we perform k-means clustering on a randomly selected set of 20,000 images, assigning identities based on clusters. We then compute the Silhouette score and ground-truth error on the same set. The reported error corresponds to the model with the best Silhouette score observed up to that point.

The hierarchical sampling allows us to address the question of how to select pairs of fragments to optimize the training speed of the network. Since we sample pairs of fragments rather than directly sampling pairs of images, we need to skew the probability of a pair of fragments being sampled to reflect the number of images they contain. More concretely, let $f_i$ be the number of images in fragment $F_i$. For negative relations, we define $f_{i,j} = f_i + f_j$ and set the probability of sampling the pair $F_i, F_j$, by their size as

$$P_s(F_i, F_j) = \frac{f_{i,j}}{\sum_{k=1}^{N-1} \sum_{l=k+1}^{N} f_{k,l}}. \tag{3}$$

For positive pairs, the probability of sampling a given fragment $f_i$ is

$$P_s(F_i) = \frac{f_i}{\sum_{j=1}^{N} f_j}. \tag{4}$$

By examining the evolution of the clusters during training (**Figure 2c**), it is clear that the learning process is not uniform, as some of the identities become separated sooner than others. **Figure 2c** top row second and third columns give us a nice illustration of this phenomenon. The images embedded in the pink rectangle of the representational space already satisfy the loss function, meaning that the negative pairwise relationships are already embedded further away than $D_{\text{neg}}$, and images that form positive pairwise relationships are already embedded closer than $D_{\text{pos}}$. Consequently, the loss function for these pairs is effectively zero, and passing them through the network will not alter the weights, merely prolonging the training process. In contrast, the separation of clusters in the orange square is incomplete, indicating that image pairs in this region still contribute to the loss function. These pairs are more pertinent, as they contain information that the network has yet to learn. To bias the sampling of image pairs towards those that still contribute to the loss function, each pair of fragments is assigned a loss score. When a pair of images is sampled for training, if the loss for that pair is not zero, the loss score for the corresponding pair of fragments is incremented by one. This score then undergoes an exponential decay of 2% per batch. More specifically, let $l_s(i,j)$ be the loss score of the pair of fragments $F_i$ and $F_j$ and $\mathcal{L}(I_{il}, I_{ik})$ the loss of the images $I_{il}$ and $I_{ik}$. If the pair $I_{il}$ and $I_{ik}$ is sampled, the loss score is updated by

$$l_s(i,j) \longleftarrow \begin{cases} (l_s(i,j) + 1)(1 - 0.02), & \text{if } \mathcal{L}(I_{il}, I_{ik}) > 0 \\ \\ l_s(i,j)(1 - 0.02), & \text{otherwise} \end{cases} \tag{5}$$

The exponential decay is always applied independently to every pair of fragments, regardless of whether the pairs of images were sampled from those fragments in the previous batch of images or not. The loss score is converted into a probability distribution over all pairs of fragments by

$$P_{l_s}(F_i, F_j) = \begin{cases} \dfrac{l_s(i,j)}{\sum_{i \neq j} l_s(i,j)}, & \text{if } i \neq j \\ \\ \dfrac{l_s(i,i)}{\sum_i l_s(i,i)}, & \text{otherwise} \end{cases} \tag{6}$$

The final probability of sampling pairs of fragments is given by

$$P(F_i, F_j) = \alpha P_s(F_i, F_j) + (1 - \alpha) P_{l_s}(F_i, F_j) \tag{7}$$

This balance between these two probabilities can be seen as an exploitation versus exploration paradigm. $P_s(F_i, F_j)$ enforces constant exploration, while $P_{l_s}(F_i, F_j)$ exploits the current state of learning by dynamically updating the sampling probability. This ensures that pairs of fragments containing unlearned knowledge are sampled more frequently, while maintaining a baseline of exploration based on fragment size. We tried several values for α and found that a value of α around 0.5 produced the best decrease of the time required to train the network (*Appendix 3—figure 6*).

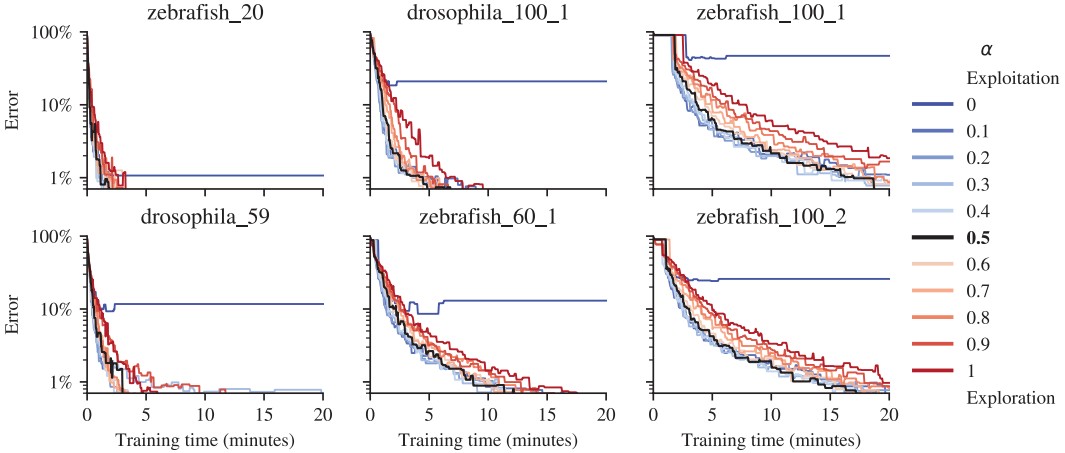

**Appendix 3—figure 6.** Exploration and exploitation comparison. Error in image identification as a function of training time for different exploration/exploitation weights α in six test videos. Every 100 training batches, we perform k-means clustering on a randomly selected set of 20,000 images, assigning identities based on clusters. We then compute the Silhouette score and ground-truth error on the same set. The reported error corresponds to the model with the best Silhouette score observed up to that point.

## Clustering and assignment

After training the network using contrastive loss, we pass all images through the network to generate their corresponding embeddings in the learned representational space. These embeddings are then grouped using mini-batch k-means clustering (*Sculley, 2010*). Each cluster ideally represents images of the same identity, as the training process has encouraged the network to place similar images close together and dissimilar ones farther apart in the embedding space. Next, we perform single-image classification, assigning each image a label based on the cluster to which its embedding belongs. Afterward, the assignment method follows two conditions. If more than half of the images in the video cannot be identified (because their predicted identities are not coherent with the video structure) and there exist global fragments, the accumulation protocol from the original idtracker.ai

is run, see 'Fallback accumulation protocol'. Otherwise, we move straight to residual identification, see 'Residual identification'.

In order to identify fragments, we not only need an identity prediction for each image but also a probability distribution over all the identities. Let $d_j(I_{ik})$ be the distance of image $I_{ik}$ to the center of cluster $j$. We define the probability of image $I_{ik}$ belonging to identity $j$ by

$$P\big(I_{ik} \text{ belongs to identity } j\big) = \frac{1/d_j(I_{ik})^7}{\sum_m 1/d_m(I_{ik})^7} \tag{8}$$

*Equation 8* is used to emphasize differences in distances between points and clusters, creating a more peaked probability distribution that clearly distinguishes closer clusters from farther ones. The exponent of seven smooths the probability distribution and reduces the influence of distant clusters, making the assignment more discriminative. In higher-dimensional spaces like the eight-dimensional space in the paper, distances are more spread out, and using a high power helps to counteract this dispersion, resulting in more confident cluster assignments.

If we are in a scenario where no global fragments exist, k-means is initialized with greedy k-means++ (*Arthur and Vassilvitskii, 2007*). Otherwise, we use the average embedding of the fragment images in one of the global fragments as initial cluster centers. This approach speeds up and stabilizes convergence, allowing us to better compare clusters as training progresses.

## Differences with previous work in contrastive/metric learning

Recent advances in representation learning have established powerful, scalable tools for learning visual features without labels. This progress has been spearheaded by contrastive methods such as SimCLR/v2 (*Chen et al., 2020a*; *Chen et al., 2020b*), SimSiam (*Chen and He, 2021*), and Momentum Contrast (MoCo) (*He et al., 2020*), and reconstruction/self-distillation techniques like Masked Autoencoders (MAE) (*He et al., 2022*) and DINOv2 (*Oquab et al., 2024*). Collectively, these frameworks have delivered state-of-the-art results in recognition and feature learning. Their success is demonstrated in specialized domains like human and primate facial recognition (*Schroff et al., 2015*; *Guo et al., 2020*), as well as in setting new standards for general visual representation quality (*Chen et al., 2020a*; *Chen et al., 2020b*; *He et al., 2020*).

The success of these models has also underpinned foundational progress in Multiple Object Tracking (MOT) via Re-Identification (ReID). Their influence is seen in the evolution of ReID-driven tracking, from early Deep Metric Learning (*Yi et al., 2014*) and DeepReID (*Li et al., 2014*) formulations to the sophisticated MOT methods used today (*Dong and Shen, 2018*; *Liang et al., 2020*; *Wang et al., 2020*; *Yang et al., 2020*).

Against this backdrop, we clarify how our formulation aligns with, and deliberately departs from, these widely adopted variants.

- **No image augmentations**. We do not apply rotations or other heavy augmentations to generate positives. Positives and negatives are instead sampled from heuristically tracked fragments: same-fragment crops provide positives, and temporally coexisting fragments provide negatives. Because crops are egocentrically pre-aligned, additional invariances are unnecessary and can suppress fine identity cues that are critical in our setting.

- **No projection head**. Unlike SimCLR-style pipelines that add a projection MLP before the contrastive loss *Chen et al., 2020a*; *Chen et al., 2020b*, we train a lightweight encoder to output a direct, low-dimensional embedding for the sole downstream task of identity clustering. Since we do not target broad transfer or multi-task robustness, a projection head intended to filter 'nuisance' features is not required.

- **No stop-gradient**. BYOL and SimSiam prevent representational collapse via stop-gradient asymmetry (*Grill et al., 2020*; *Chen and He, 2021*). In our case, fragments naturally yield abundant, high-quality negatives; balanced positive/negative batches and a loss-aware pair sampler maintain training signal without asymmetry. This preserves simplicity while addressing collapse risks that arise when negatives are scarce.

- **Euclidean margins rather than cosine similarity**. While many contrastive methods operate on L2-normalized features with cosine similarity, we optimize Euclidean distances with explicit thresholds $D_{\text{pos}}$ and $D_{\text{neg}}$. These margins give direct control over intra- and inter-cluster spacing

and align naturally with downstream k-means assignment used to form identities (as opposed to cosine-margin classifiers).

- **Novel pairs sampling strategy**. Our fragment-level, loss-aware sampling is related to online hard example mining (OHEM) and hard negative sampling for contrastive learning (*Shrivastava et al., 2016*; *Robinson et al., 2021*), but it differs in two key ways that are specific to our setting. First, selection happens over *fragments* defined by temporal co-existence rather than over labeled instances or memory-bank entries, and hardness is estimated online from a decayed record of recent non-zero loss, without curated queues or global rankers. Second, the convex mixture $P = \alpha P_s + (1 - \alpha) P_{l_s}$ makes the policy explicitly *exploration–exploitation*: $P_s$ guarantees coverage of the fragment graph (preventing sampling starvation), while $P_{l_s}$ focuses on unresolved regions of the embedding. This bandit-style view clarifies our empirical findings: pure exploitation ($\alpha = 0$) collapses onto a few regions (catastrophic forgetting), whereas a mid-range α maintains global progress while accelerating convergence.

## Stopping criteria

Stopping network training using the loss function directly can be highly variable, as different video conditions, the number of individuals, and the sampling method significantly influence this value. To circumvent this, we use the Silhouette score (SS) (*Rousseeuw, 1987*) of the clusters of the embedded images. Let $d(I, J)$ be the Euclidean distance between the embeddings of image $I$ and $J$, for each image $I$, in cluster $C_a$ we compute the mean intra-cluster distance

$$a(I) = \frac{1}{|C_a| - 1} \sum_{J \in C_a, J \neq I} d(I, J) \tag{9}$$

and the mean nearest-cluster distance

$$b(I) = \min_{a \neq b} \frac{1}{|C_b|} \sum_{J \in C_b} d(I, J) \tag{10}$$

The SS is given by

$$SS = \frac{1}{\text{number of images}} \sum_{I} \frac{b(I) - a(I)}{\max\{b(I), a(I)\}} \tag{11}$$

To determine when to stop training, every $m$ batches we compute the SS by clustering the embeddings of a random sample of the images in the video, generating also a checkpoint of the model. $m$ was set to be the maximum between 100 and the number of animals in a video times 5. We stop training if: (1) there have been 30 consecutive SS evaluations without any improvement (patience of 30), or (2) there have been 2 consecutive SS evaluations without any improvement but the SS already achieved a value of 0.91. After stopping the training, the checkpoint model with the highest SS is chosen. A threshold of 0.91 was validated empirically (*Figure 2d and e*). The number of images used for the computation of the SS is 1000 times the number of animals.

## Fallback accumulation protocol

The contrastive approach is considered to fail when it can confidently identify only less than half of the images in the video (with `CONTRASTIVE_MIN_ACCUMULATION=0.5`, but users can modify this value). In such cases, and only if the video contains global fragments, protocol 2 from the original idtracker.ai is applied. This is a fallback tracking mechanism and, notably, the system never had to apply it in any of the videos of our benchmark.

The way protocol 2 from the original idtracker.ai is applied in this scenario is as follows. Fragments identified by the contrastive algorithm serve as an initial labeled dataset, effectively creating a synthetic initial global fragment in terms of the original idtracker.ai. Using this dataset, the small CNN from the original idtracker.ai is taken as the new identification network and trained. The trained model then predicts the identities of the remaining unidentified fragments in the video. Then, quality checks are applied to filter out potentially incorrect identity assignments (see Section B.2.4, *Cascade of training/identification protocols*, in Appendix B of *Romero-Ferrero et al., 2019*). Only fragments passing these checks are accumulated into the training dataset. The identification network is then

retrained with the expanded dataset, and the prediction-checks-accumulation cycle is repeated until either (i) 99.95% of the images are assigned or (ii) no further fragments pass the quality criteria. In either case, the pipeline proceeds with the 'Residual identification' step.

## Residual identification

Residual identification addresses fragments that did not pass the quality checks or were excluded due to short length. The identification network (ResNet or the small CNN, depending on whether the contrastive algorithm worked well or failed, respectively), predicts the identities of these fragments, and a probabilistic method that accounts for the coexistence of already assigned fragments is applied. Assignments are made iteratively in descending order of confidence, with the highest-confidence fragments resolved first (see Section B.2.5, *Residual identification*, in Appendix B of *Romero-Ferrero et al., 2019*).

After the residual identification, the remaining processing stages are the same as in the original idtracker.ai. These include post-processing to correct impossible trajectory jumps, interpolation through crossing events, and the generation of the final output trajectories.

## Appendix 4

### Example workflow

A typical workflow with idtracker.ai begins when a user records a video of a multi-animal experiment under laboratory conditions. The process starts by launching the Segmentation app (***Appendix 4— figure 1***) using the command idtrackerai (see full documentation at https://idtracker.ai/latest/user_guide/segmentation_app.html). This application guides users through setting the essential tracking parameters, such as the number of animals, background subtraction options, and filters for non-animal objects (using regions of interest and minimum blob size). Once configured, tracking runs automatically. Additionally, the state of an already trained ResNet from another video can be used to initialize the current model's state using the parameter `knowledge_transfer_folder`, speeding up the training process. Users can save all these parameters to a file and execute tracking from the terminal or a script with `idtrackerai --load parameters.toml --track`, which is useful for remote or batch processing.

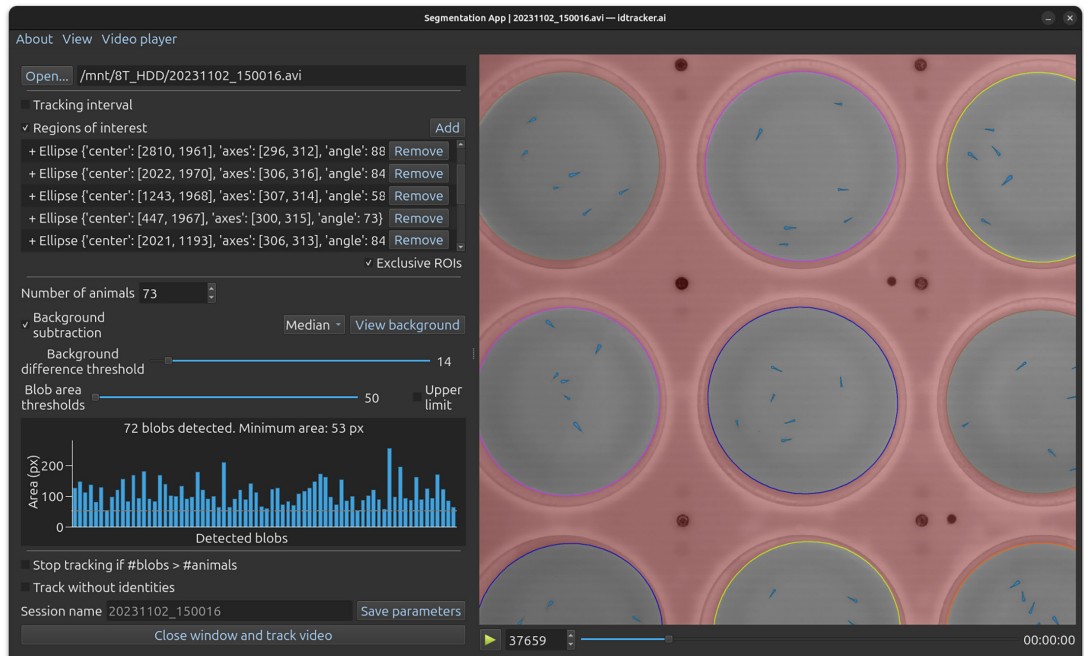

**Appendix 4—figure 1.** Segmentation GUI. Enables users to set the basic parameters required for running idtracker.ai.

After tracking, the software outputs $x$ and $y$ coordinates for each identified animal in every video frame. These trajectories are saved in multiple formats (CSV, HDF5, Numpy) and include metadata such as

- Video properties (height, width, frame rate, average blob size per identity)
- An overall estimate of tracking accuracy
- Identification probabilities for each animal in every frame
- The Silhouette score achieved during training

To assess the robustness of the learned representation space, users can generate a t-SNE plot of the embeddings from a sample of animal images by running the command `idtrackerai_inspect_clusters`, as shown in ***Figure 2c*** (see documentation at https://idtracker.ai/latest/user_guide/data_analysis.html).

Although idtracker.ai achieves over 99% IDF1 score on well-recorded videos, some frames may still contain missing or mislabeled animals. To address this, users can launch the Validator app with `idtrackerai_validate` to manually review and correct tracking results (see documentation at https://idtracker.ai/latest/user_guide/validator.html). This tool allows to navigate through video frames, inspect the tracked positions and metadata, and detect and correct errors using integrated plugins (***Appendix 4—figure 2***).

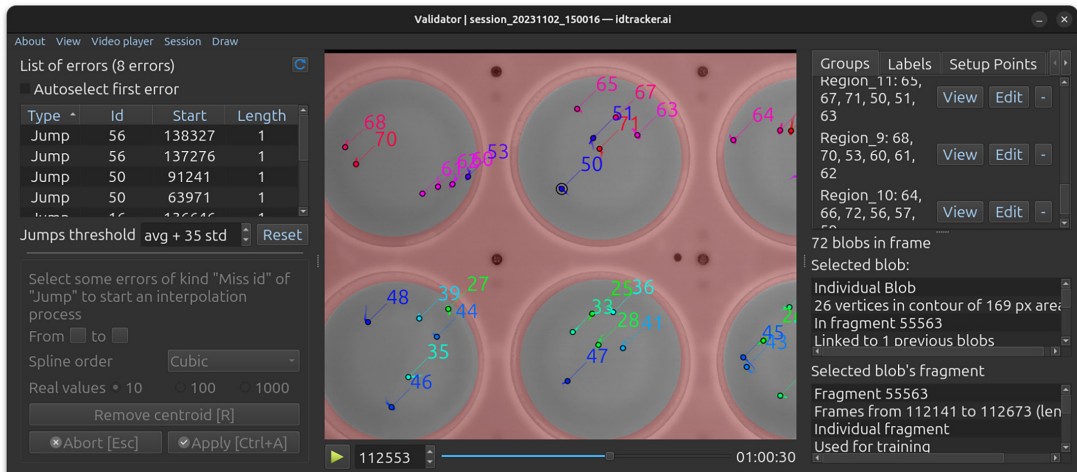

**Appendix 4—figure 2.** Validator GUI. Enables users to inspect tracking results, correct errors, and access additional tools.

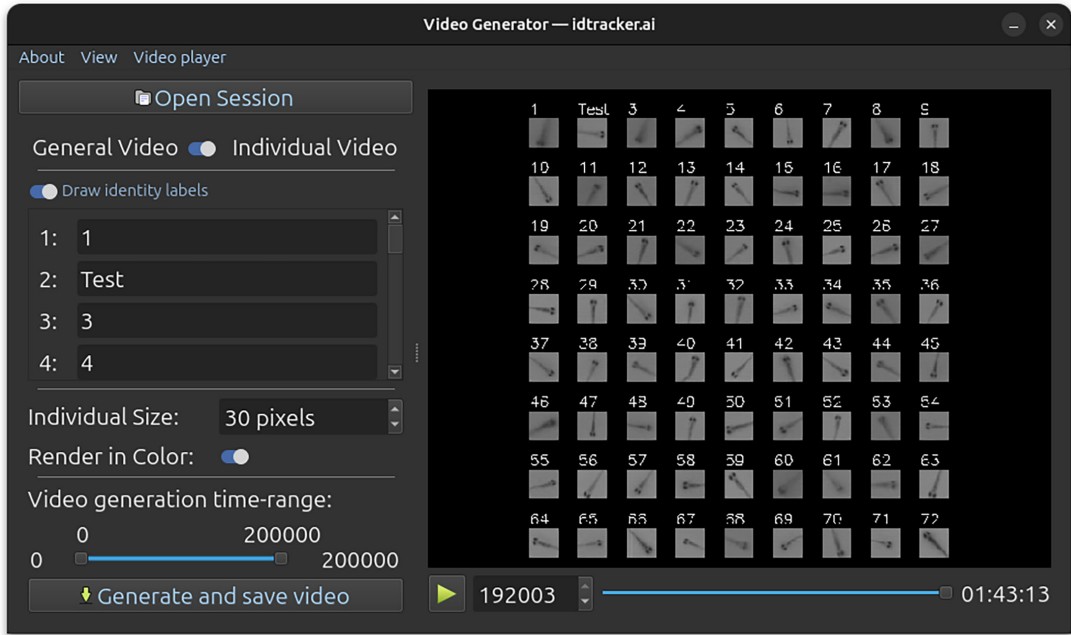

**Appendix 4—figure 3.** Video Generator GUI. Allows users to define parameters for general and individual video generation.

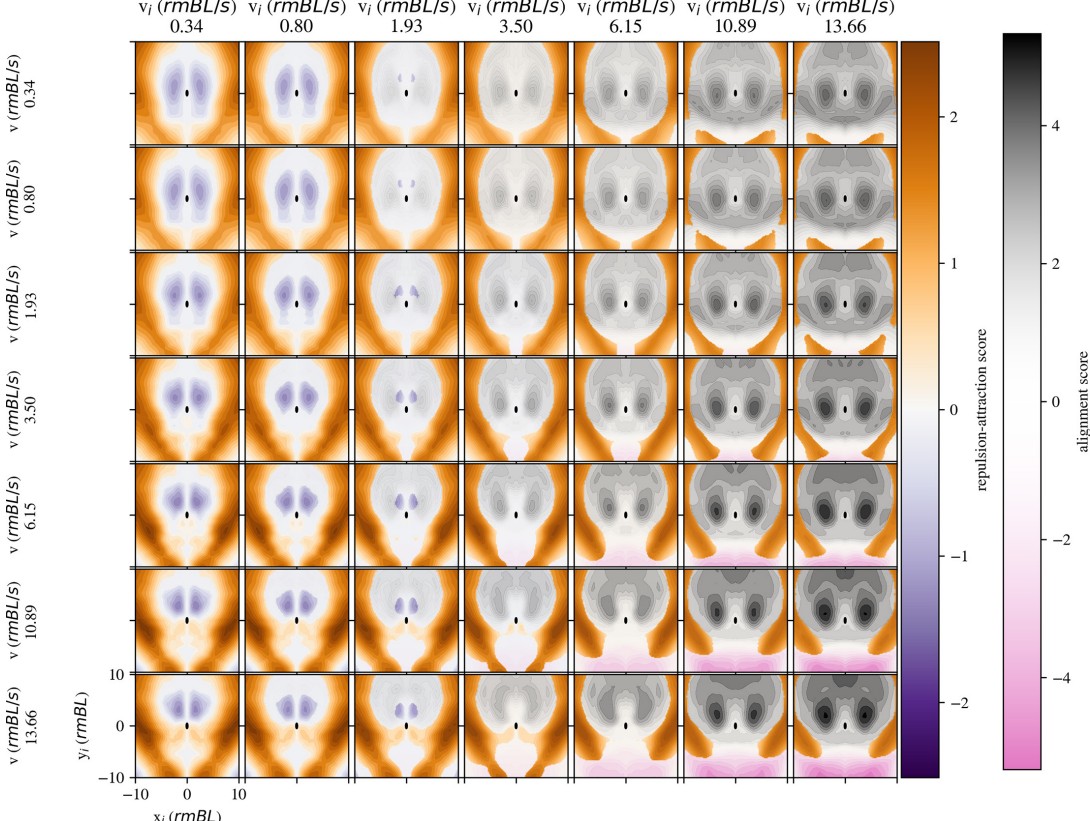

**Appendix 4—figure 4.** SocialNet output example showing learned attraction-repulsion and alignment areas for social interactions around the focal animal.

To share the tracking results, the command `idtrackerai_video` launches a graphical app (*Appendix 4—figure 3*) to generate videos with animal trajectories and optional labels overlaid on the original footage (see documentation at https://idtracker.ai/latest/user_guide/video_generators. html). It can also produce individual videos for each animal, showing only its cropped region over time. These individual videos can be used as input for pose estimation tools such as DeepLabCut (*Lauer et al., 2022*), SLEAP (*Pereira et al., 2022*), MARS (*Segalin et al., 2021*), ADPT (*Tang et al., 2025*), and Lightning Pose (*Biderman et al., 2024*).

When experiments involve the same animals across multiple sessions, users can match identities between these sessions using `idmatcher.ai` (originally introduced in *Romero-Ferrero et al., 2023* and now integrated into the idtracker.ai ecosystem; see documentation at https://idtracker.ai/latest/ user_guide/idmatcherai.html and Appendix 5). This tool ensures consistent identity labeling across independent recordings, supporting multi-session studies.

The resulting trajectories can be loaded and analyzed in any programming language. For this purpose, we developed the Python package *trajectorytools*. It offers utilities for the analysis of 2D trajectory data, basic movements metrics and spatial relationships between the animals, at https:// idtracker.ai/latest/user_guide/data_analysis.html#trajectorytools. In the same URL, we also provide a set of Jupyter Notebooks to facilitate its usage providing analysis examples.

Additionally, SocialNet (*Heras et al., 2019*), a model of collective behavior, can be seamlessly applied to the trajectories generated by idtracker.ai to extract social interaction rules, as illustrated in *Appendix 4—figure 4* (see documentation at https://idtracker.ai/latest/user_guide/socialnet.html).

## Appendix 5

### Validity of idmatcher.ai in the new idtracker.ai

idmatcher.ai, first introduced in *Romero-Ferrero et al., 2023*, is an integrated tool in idtracker.ai to match identities across videos of the same animals. After tracking two separate videos with an arbitrary order of the identity labels, the identification networks from both are used to cross-identify animal images from the other video. This generates a confusion matrix from which final identities are obtained by solving the assignment problem with a modified Jonker-Volgenant algorithm (*Crouse, 2016*).

The reliability of idmatcher.ai was already proven for the networks of the previous versions of idtracker.ai (v5 and previous). To confirm its performance also in v6, several videos from the benchmark are split into two non-overlapping parts with a 200 frames gap between them. Both parts are tracked independently, and idmatcher.ai is then applied to match identities between them. This test (with the videos z_20, z_60_1, z_100_2, z_100_3, z_80_1, z_80_2, z_80_3, z_100_1, d_60, d_72, d_80, d_100_1, d_100_2, and d_100_3) presents an average image-level accuracy of 89%, enough for a perfect identity matching with zero errors (see *Appendix 5—figure 1* for a specific example with *drosophila_80*).

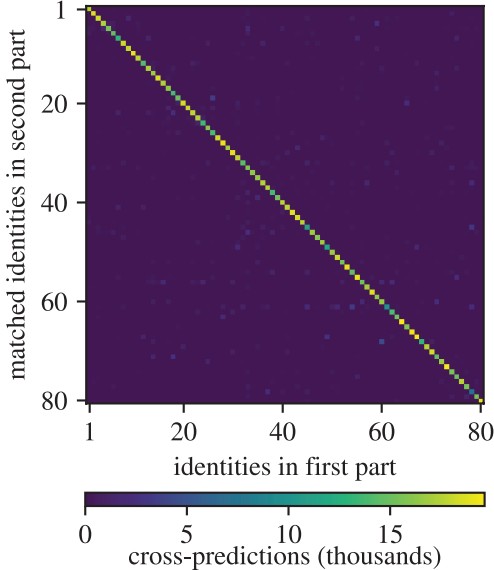

**Appendix 5—figure 1.** Confusion matrix between the two parts of *drosophila_80* in v6 of idmatcher.ai. It contains predictions both from the network trained in the first part with images from the second one and the other way around. In this example, the image-level accuracy is 82.9%, enough for a 100% accurate identity matching.

