## [Editor Report · eLife Assessment]

This **important** study introduces an advance in multi-animal tracking by reframing identity assignment as a self-supervised contrastive representation learning problem. It eliminates the need for segments of video where all animals are simultaneously visible and individually identifiable, and significantly improves tracking speed, accuracy, and robustness with respect to occlusion. This innovation, which is supported through **compelling** evidence, has implications beyond animal tracking, potentially connecting with advances in behavioral analysis and computer vision.

---

## [Referee Report · Reviewer #3 (Public review)]

Summary:

The authors propose a new version of idTracker.ai for animal tracking. Specifically, they apply contrastive learning to embed cropped images of animals into a feature space where clusters correspond to individual animal identities. By doing this, they address the requirement for so-called global fragments - segments of the video, in which all entities are visible/detected at the same time. In general, the new method reduces the long tracking times from the previous versions, while also increasing the average accuracy of assigning the identity labels.

Comments on revisions:

I have no additional comments, the authors have responded to all the points I raised previously.

---

## [Author Response]

The following is the authors’ response to the previous reviews

**eLife Assessment**
This important study introduces an advance in multi-animal tracking by reframing identity assignment as a self-supervised contrastive representation learning problem. It eliminates the need for segments of video where all animals are simultaneously visible and individually identifiable, and significantly improves tracking speed, accuracy, and robustness with respect to occlusion. This innovation has implications beyond animal tracking, potentially connecting with advances in behavioral analysis and computer vision. The strength of support for these advances is compelling overall, although there were some remaining minor methodological concerns.

To tackle “minor methodological concerns” mentioned in the Editorial assessment and Reviewer 3, the new version of the manuscript includes the following changes:

a) The new ms does not anymore use the word “accuracy” but “IDF1 scores”. See, for example, Lines 46, 161, 176, and 522 for our new wording as “IDF1 scores”.

b) Instead of comparing softwares using mean accuracy over the benchmark, Reviewer 3 proposes to use medians or even boxplots. We now provide boxplot results with mean, median, percentiles and outliers (Figure 1- figure Supplement 2).

Additionally, we also include in the text the other recommendations from Reviewer 3:

a) We now more explicitly describe the problems of the original idtracker.ai v4 in the benchmark (lines 66-68). Around half of the videos had a high accuracy in the original dtracker.ai (v4) but the other half of the videos had lower accuracies (Figure 1a, blue). The new idtracker.ai has high accuracy values for all the videos (Figure 1a, magenta).

Also, the videos with high accuracy in the old idtracker.ai had very long tracking times (Figure 1b, blue) and the new version does not (Figure 1b, magenta). So the benchmark allows us to distinguish the new idtracker.ai as having a better accuracy for all videos and lower tracking times, making it a much more practical system than previous ones.

b) We further clarified the occlusion experiment (lines 188-190 and 277-290).

c) We explain why we measure accuracies with and without animal crossings (lines 49-62).

d) We added a Discussion section (lines 223-244).

We believe the new version has clarified the minor methodological concerns.

**Reviewer #3 (Public review):**
The authors have reorganized and rewritten a substantial portion of their manuscript, which has improved the overall clarity and structure to some extent. In particular, omitting the different protocols enhanced readability. However, all technical details are now in appendix which is now referred to more frequently in the manuscript, which was already the case in the initial submission. These frequent references to the appendix - and even to appendices from previous versions - make it difficult to read and fully understand the method and the evaluations in detail. A more self-contained description of the method within the main text would be highly appreciated.

In the new ms, we have reduced the references to the appendix by having a more detailed explanation in one place, lines 49-62.

Furthermore, the authors state that they changed their evaluation metric from accuracy to IDF1. However, throughout the manuscript they continue to refer to "accuracy" when evaluating and comparing results. It is unclear which accuracy metric was used or whether the authors are confusing the two metrics. This point needs clarification, as IDF1 is not an "accuracy" measure but rather an F1-score over identity assignments.

We thank the reviewer for noticing this. Following this recommendation, we changed how we refer to the accuracy measure with “IDF1 score” in the entire ms. See, for example, lines 46, 161, 176, and 522.

The authors compare the speedups of the new version with those of the previous ones by taking the average. However, it appears that there are striking outliers in the tracking performance data (see Supplementary Table 1-4). Therefore, using the average may not be the most appropriate way to compare. The authors should consider using the median or providing more detailed statistics (e.g., boxplots) to better illustrate the distributions.

We thank the reviewer for asking for more detailed statistics. We added the requested box plot in Figure 1- figure Supplement 2 to provide more complete statistics in the comparison.

The authors did not provide any conclusion or discussion section. Including a concise conclusion that summarizes the main findings and their implications would help to convey the message of the manuscript.

We added a Discussion section in lines 223-244.

The authors report an improvement in the mean accuracy across all benchmarks from 99.49% to 99.82% (with crossings). While this represents a slight improvement, the datasets used for benchmarking seem relatively simple and already largely "solved". Therefore, the impact of this work on the field may be limited. It would be more informative to evaluate the method on more challenging datasets that include frequent occlusions, crossings, or animals with similar appearances.

Around half of the videos also had a very high accuracy in the original dtracker.ai (v4) but the other half of the videos had lower accuracies (Figure 1a, blue). For example, we found IDF1 scores of 94.47% for a video of 100 zebrafish with thousands of crossings (*z_100_1*), 93.77% for a video of 4 mice (*m_4_2*) and 69.66% for a video of 100 flies (*d_100_3*). The new idtracker.ai has high accuracy values for all the videos (Figure 1a, magenta).

Importantly, the tracking times for the majority of videos was very high in the original idtracker.ai (Figure 1b, blue), making the use of the tracking system limited in practice. The new system manages both a high accuracy in all videos (Figure 1a, magenta) and much lower tracking times (Figure 1b, magenta), making it a much more practical system..

We have added a sentence of the limitations of the original idtracker.ai as obtained from the benchmark, lines 66-68.

The accuracy reported in the main text is "without crossings" - this seems like incomplete evaluation, especially that tracking objects that do not cross seems a straightforward task. Information is missing why crossings are a problem and are dealt with separately.

We have now added an explanation on why we measure accuracy without crossings and why we separated it from the accuracy for all the trajectory in lines 49-62. The reason is that the identification algorithm being presented in this ms only identifies animal images outside the crossings. This algorithm makes robust animal identifications through the video despite the thousands of animal crossings typically existing in each of our videos used in the benchmark. It is a second algorithm (that hasn’t changed since the first idTracker in 2014) the one that assigns animal positions during crossings once the first algorithm has made animal identifications before and after the crossings.

There are several videos with a much lower tracking accuracy, explaining what the challenges of these videos are and why the method fails in such cases would help to understand the method's usability and weak points.

Some videos had low accuracy on previous versions (Figure 1a, blue), but the new idtracker.ai has high accuracy in all of them (Figure 1a, magenta).

**Reviewer #3 (Recommendations for the authors):**
(1) As described before, the authors claim to use IDF1 as their metric in the whole manuscript (lines 414-436) but only refer to accuracy when presenting the results. It is not clear, whether accuracy was used as a metric instead of IDF1 or the authors are confusing these metrics.

Following this recommendation, we replaced “accuracy” with “IDF1 score” , see lines 46, 161, 176, and 522.

(2) In the introduction, a brief explanation why crossings need to be dealt with separately would help to understand the logic of the method design.

We added such an explanation in lines 49-62.

(3) Figure 3: We asked about how the tracking accuracy is being assessed with occlusions. The authors responded with that only the GT points inside the ROI are taken into account when computing the accuracy. Does this mean, that the occluded blobs are still part of the CNN training and the clustering? This questions the purpose of this experiment, since the accuracy performance would therefore only change, if the errors, that their approach is doing either way, are outside the ROI and, therefore, not part of the metric evaluation.

The occluded blobs are not part of any training because they are erased from the video, they do not exist. We made this more clear in lines 188-190 and 277-290.

(4) Figure 1: The fact that datasets are connected with a line is misleading - there is no connection between the data along the x-axis. A line plot is not an appropriate way to present these results.

The new ms clarifies that the lines are for ease of visualization, see last line in the caption of Figure 1.

(5) Lines 38-39: It is not clear how the CNN can be pretrained for the entire video if there are no global segments or only short ones. Here, the distinction between "no segments", "only short segments" and "pretraining on the entire video" is not explained.

This pretraining protocol is not used in the version of the software we present, so details of this are not as relevant.

(6) Figure 2a: The authors are showing "individual fragments" and individual fragments in a global fragment." However, it seems there are a few blue borders missing. In the text (l. 73-79), they note, that they are displaying them as "examples" but the absence of correct blue borders is confusing.

In the new ms, we have replaced the label “Individual fragments in a global fragment” with “Individual fragments in an example global fragment” in the legend of Figure 2.

(7) Lines 61-63, 148-151, and 162-164: Could the authors clarify why they used the average instead of median when comparing the speedups of the new version and the old ones?

We thank the reviewer for asking for more detailed statistics. We added the requested box plot in Figure 1- figure Supplement 2 to provide more complete statistics in the comparison of accuracies and tracking times for old and new systems.

(8) Lines 140-144: The post-processing steps are not clear. The authors should rather state clearly which processes of the old versions they are using. Then the authors could shortly explain them.

We removed this paragraph and explained in more detail in lines 49-62 which parts of the software are new and which ones are not.

(9) Lines 239-251: Here, the authors are clarifying on a section 1-2 pages before. This information should be directly in that section instead.

Following this recommendation, we clarified the occlusion experiment in the main text (lines 188-191) to make it more self-contained. Still, the flow of the main text is better with some details in Methods.

(10) Line 38: It is not clear how the CNN can be pretrained for the entire video if there are no global segments or only short ones. Here, the distinction between "no segments""only short segments" and "pretraining on the entire video" is a bit misleading/underexplained.

See number 5.

(11) Figure 2a: The authors are showing "individual fragments" and individual fragments in a global fragment." However, it seems there are a few blue borders missing. In the text (l. 73-79), they note, that they are displaying them as "examples" but the absence of correct blue borders is confusing.

See number 6.

(12) Figure 2c and line 115-118: "Batches" itself is not meaningful without any information of the batch size. The authors should rather depict the batch size and then the number of epochs. The Figure 2 contains the info 400 positive and 400 negative pairs of images per batch. However, there is no information about the total number of images.Furthermore, these metrics are inappropriate here, since training is carried out from scratch (or already pre-trained) for every new video, each video has different number of animals, different number of images.

Following this recommendation, we clarified the number of images in each batch (Figure 1c caption and lines 134-138), why we do not work with epochs (lines 700-702), and the idea that the clusters in Figure 2 represent an example and the number of batches needed for the clusters to form depends on the video details.

Appendix 1-figure 1: why do the methods fail? It looks that for certain videos the method is fairly unreliable. What is the reason for the methods to crash and how to avoid this?

Those failures are only for the old idtracker.ai and Trex, not for the method presented here. Our new contrastive algorithm does not fail in any of the videos in the benchmark.

We thank the reviewer for the detailed suggestions. We believe we have incorporated all of them in the new version of the ms.